Preliminary study on Bioassay of Capparis spinosa L. seed extract and seed germination

Wang Min
Yuan Xiaolu
Xu Liping xuliping70@126.com
College of Biological and Geographical Science, Institute of Resources and Ecology, Yili Normal University , Yining, Xinjiang , China
Shrestha Jiban
Electronic publication date: 2023 Mar 14
Publication date: 2023
Volume: 11
Electronic Location ID: e15082
Received 2022 Oct 11; Accepted 2023 Feb 25
Copyright: © 2023 Wang et al.
Copyright year: 2023
Copyright holder: Wang et al.
License: This is an open access article distributed under the terms of the Creative Commons Attribution License, which permits unrestricted use, distribution, reproduction and adaptation in any medium and for any purpose provided that it is properly attributed. For attribution, the original author(s), title, publication source (PeerJ) and either DOI or URL of the article must be cited.
License URL: https://creativecommons.org/licenses/by/4.0/

Keywords: Capparis spinosa, Germination inhibitor, Chinese cabbage, Seed extract

Funding: Natural Science Foundation of Science and Technology 2020D01C267 This work was supported by the Natural Science Foundation of Science and Technology Department of Xinjiang Uygur Autonomous Region (2020D01C267): Study on seedling characteristics and drought tolerance response mechanism of Capparis spinosa L. from different provenances. The funders had a role in study design, data collection and analysis, decision to publish, or preparation of the manuscript.

==============================
The present study explored the germination inhibitors present in the seeds of Capparis spinosa L., a plant species that is known for its ecological significance in preventing wind erosion and fixing sand in desertified areas. Additionally, its roots, leaves, and fruits possess medicinal properties, and are used to treat a range of ailments such as rheumatism, tumors, and diabetes. However, the plant’s low germination rate under natural conditions is a major limitation. We aimed to improve the germination of C. spinosa seeds by investigating the effects of various infusions of caper seeds on the germination and seedling growth of Chinese cabbage seeds. A range of chemical reagents, hormonal immersions, and sand storage treatments were used to determine the differences in the germination rate of C. spinosa seeds. Our results revealed that among the various water extract concentrations tested, 100% water extract exhibited the strongest inhibitory effect on the germination and growth of the cabbage seeds, with a germination rate of (70.00 ± 0.09)%. Furthermore, the inhibitory effects on the germination and growth of cabbage seeds were found to be strongest when treated with the extract solution 1, yielding a germination rate of (83.33 ± 0.02)%. Notably, the leaves of Chinese cabbage seedlings turned yellow-green and yellow after treatment with the extract solution. These findings highlight the potential inhibitory effects of C. spinosa seed extracts on seed germination and growth and suggest that further research is needed to better understand the underlying mechanisms. The results of the germination experiment with methanol extract showed a sharp decline in the germination rate of Chinese cabbage seeds treated with 50% methanol extract, to (4.67 ± 0.02)%. These findings indicate the presence of germination-inhibiting substances in caper seeds. The highest germination potential was observed when the caper seeds were soaked in 30% PEG, reaching 35.00%. The highest germination rate, 19.33%, was observed when the seeds were soaked in 250 mg/L GA3 and 25 mmol/L NaCl. These results suggest that the germination inhibitor present in caper seeds affects the germination of cabbage seeds as well. The highest germination rate was achieved when the seeds were soaked with gibberellin. It is hoped that the research on the germination-inhibiting substances in caper seeds will provide a scientific foundation for improving and refining the artificial propagation and cultivation methods of this species.

Introduction

Capparis spinosa L. is a deciduous subshrub that can grow up to 30–50 cm in height, and it is characterized by its deep, sturdy root system and branches that can reach up to 3 m in length (Foschi et al., 2020; Yang et al., 2008). This plant is widely distributed in Mediterranean coastal countries and the Middle East, and can also be found in regions of China such as Gansu, Xinjiang, and Tibet (Inocencio et al., 2002; Wang, Zhang & Yin, 2016; Fici, 2001). C. spinosa has a range of medicinal and ecological benefits, including anti-rheumatic and liver-protective properties, as well as the ability to reduce soil erosion and resist wind and sand (Zhang & Hai, 2002; Rahnavard & Razavi, 2016). Additionally, the plant’s seeds can be consumed as a pickle or condiment (Mazandarani, Borhani & Fathiazad, 2014). The seeds have a high nutritional value, containing 26% fiber and 19–22% protein, and they serve as a significant source of oil (Biouki, Khajahhosseini & Rad, 2020). However, in China, caper plants are mainly found in the wild and artificial breeding is relatively rare. The low natural germination rate and scarcity of existing populations has contributed to the plant’s relative scarcity and limited resources (Levizou, Drillas & Kyparissis, 2004; Khaninejad, Arefi & Kaf, 2012; Juan et al., 2020; Baskin & Baskin, 2014).

Seed dormancy constitutes a significant factor contributing to the low germination rate of capers. The causes of plant seed dormancy can be classified into two broad categories: external environmental factors, such as air, moisture, and temperature, and intrinsic seed-related factors. Intrinsic factors encompass the presence of germination-inhibiting substances within the species, physical barriers in the seed coat, and morphological and physiological immaturity of the embryo (Guo, 2016). Their presence obstructs the metabolic connection between seed physiological activities, thereby impacting the germination process (Li et al., 2011). Seed germination inhibitors are substances that can impede or delay the germination of both inter- and intraspecific seeds, which are classified into endogenous (organic acids, abscisic acid, aldehydes, phenols, etc. produced within the seed coat or other parts of the seed) and exogenous (produced by other plants) categories. Among these, endogenous inhibitory substances are more prevalent and exert a greater impact on seed germination (Li et al., 2020; Yan et al., 2014). Their presence obstructs the metabolic connection between seed physiological activities, thereby impacting the germination process (Jia et al., 2021). Studies have demonstrated that GA3 can significantly reduce the concentration of germination inhibitors and enhance seed germination (Guan, 1986). A sand storage treatment has also been shown to improve air permeability of thick seed coats and alleviate seed dormancy to some extent (Sun, Guo & Wei, 2012). Another factor contributing to dormancy is the degradation of the membrane system, leading to changes in cell membrane permeability during storage and inducing a dormant state in seeds (Peng et al., 2016; Zhang, Chen & Zhang, 2010). FeSO4, KNO3, and PEG-6000 have been shown to regulate membrane permeability, thus breaking seed dormancy and elevating germination rate (Meng & Gao, 2008; Lin et al., 2014; Niu et al., 2018; Chen et al., 2017). In this study, we aimed initially to examine the germination inhibitors of capers by exposing seeds to FeSO4, KNO3, PEG-6000, NaCl, and GA3 solutions and through sand storage methods to break dormancy and increase germination rate. Additionally, we investigated the impact of methanolic and aqueous extracts of the caper plant on the germination of Chinese cabbage seeds, contributing to a scientific understanding of this resource for sustainable development and utilization.

Materials and Methods

The ripe fruits of Capparis spinosa L. were harvested from Karayagaqi Township in Yining County (44°10′N, 81°52′E, 1101.37 masl) in late July 2021. The caper seeds were stripped from the fruit and washed and stored at −20 °C in a refrigerator. The Chinese cabbage seeds were procured from Cangzhou Jinke Lifeng Seedling Co., Ltd. and stored at room temperature.

Seed morphology observation

The caper seeds were dissected by making a longitudinal cut along the umbilical point using a scalpel. The morphological structure of these seeds was then observed under a stereomicroscope.

Determination of water absorption rate of caper seeds

Two groups were established, an experimental group and a control group. In the experimental group, 50 seeds were subjected to sandpaper abrasion to break the seed coat and this procedure was repeated three times, totaling 150 seeds. Meanwhile, the seeds in the control group were left unaltered with their seed coats intact and replicated three times. Both groups of seeds were weighed and then soaked simultaneously in distilled water. At 2-h intervals, the seeds were drained of surface water using filter paper and weighed. The weight of the seeds was recorded every 2 h, until a point where no further weight gain was observed (Zhao et al., 2016).

Waterabsorptionrate=wetweight-dryweightdryweight×100%

Treatment of cabbage seeds with water and methanol extracts from caper seeds

Caper seeds (2 g) underwent three successive infusions, each performed with 20 mL of sterile distilled water and a 24-h soak time. The resulting extracts were obtained by filtration through filter paper, yielding extracts 1, 2, and 3. Five milliliters of each infusion was used to culture cabbage seeds (50 seeds per infusion treatment), which were incubated in an artificial climate incubator with a 16-h light regime and a temperature of 25 °C. The experiment was conducted in triplicate.

Caper seeds (2 g) were subjected to two extraction methods: one with sterile distilled water (15 mL) and the other with 80% methanol (15 mL). The seeds were ground into a powder and soaked for 24 h at 25 °C prior to filtration, yielding extracts of 20 mL each. These extracts were then diluted to concentrations of 25%, 50%, 75%, and 100% of their original strength, respectively, to form a series of dilutions. The distilled water served as a control for the first extraction method, while the 80% methanol served as a control for the second extraction method.

The experiment involved two distinct extracts, each of which was prepared at four different concentrations. A total of 150 cabbage seeds were employed, with 50 seeds per concentration and three replicates per concentration. The seeds were washed, placed on Petri dishes lined with two layers of gauze (8 × 8 cm), and treated with 5 mL of the various extract concentrations. A control group was also included, resulting in a total of 30 Petri dishes used. The dishes were incubated in an artificial climate chamber with a 16-h photoperiod and a temperature of 25 °C (Luo, 2015). Germination was assessed on the 3rd day, and the root and shoot lengths of the cabbage seedlings were measured on the 5th day, with data collected 30 times.

Release of caper seeds dormancy

Treatment of seeds with chemical agents

Abraded caper seeds were subjected to a light sanding to thin their seed coat and enhance seed germination potential. The seeds underwent sterilization by soaking in 0.5% sodium hypochlorite solution for 15 min, followed by three washes with sterile water. Six concentration gradients of FeSO4 (0.1%, 0.3%, 0.5%, 0.7%, 0.9%, and 1.1%), KNO3 (2%, 3%, 3.5%, 4%, 4.5%, and 5%), PEG (10%, 15%, 20%, 25%, 30%, and 35%), and NaCl (10, 25, 50, 100, 150, and 200 mmol/L) were used to soak the caper seeds for 24 h at each gradient. The treatment was repeated three times, with 50 seeds per repetition.

The soaked seeds were then placed in Petri dishes lined with two layers of gauze (8 × 8 cm) and placed in an artificial climate incubator, with 14 h of light per day and a temperature of 30 °C during the day and 25 °C at night (Fang, Ye & Zhu, 2017; Lu, 2012; Luo et al., 2014; Zhang et al., 2009; Zheng et al., 2020).

Treatment of seeds with exogenous hormones

The GA3 concentration was varied to six levels: 175, 200, 225, 250, 275 and 300 mg/L. The seed treatment protocol and soaking duration followed the same methodology as described in the previous chemical treatment (Luo et al., 2014).

Seeds for sand storage

The seeds and sand were disinfected with 0.5% sodium hypochlorite to ensure their sterilization. Subsequently, 50 seeds were placed in pockets of gauze and laid flat on top of the sand. The humidity of the sand was controlled to approximately 40% by carefully manipulating it to form a ball, and loosening it when it became too dense. The seeds were subjected to a temperature of 4 °C or room temperature (25 °C) for 1, 7, 14, and 21 days. At the end of each storage period, the seeds were removed, washed with water, and placed in Petri dishes lined with gauze. The dishes were then incubated (Du, Li & Xue, 2013; Xue et al., 2019; Liu et al., 2022).

Determination of germination indicators

Caper seed: The germination potential of the caper seeds was evaluated on the 7th day by counting the number of seeds that had sprouted. The germination rate was then determined on the 20th day by counting the number of seeds that had fully developed cotyledons. These were used as the criterion for successful germination.

Germinationalpotential=numberofnormalgerminatedseedswithin7dnumberoftestedseeds×100%

Germinationrate=numberofnormalgerminatedseedswithin20dnumberoftestedseeds×100%

Chinese cabbage seed: The germination rate of Chinese cabbage was counted on the 3rd day.

Germinationrate=numberofnormalgerminatedseedswithin3dnumberoftestedseeds×100%

Observational and statistical methods

In order to make a fair comparison, the control group was set up with seeds that were treated with only sterile distilled water or normal sand storage without any other treatments. The results of the treatments were then compared to the control group to see if there was a significant difference in the germination rate, germination potential, root length, and shoot length. The ANOVA method is a statistical tool that allows for the analysis of multiple variables and the determination of any significant differences between the groups. The results of the ANOVA analysis will help determine if the treatments had a significant effect on the germination rate, germination potential, root length, and shoot length of the seeds.

Results

Seed morphological characteristics

The stereo microscope was employed to examine the morphology of caper seeds (Figs. 1A and 1B). The seeds that had not developed fully exhibited a dull coloration and the kernels were observed to be desiccated (Fig. 1A). In contrast, mature seeds displayed a creamy white hue and had plump kernels (Fig. 1B).

Figure 1 Abortive seed (A) and mature seed profile (B) (magnification: 10 × 2).

Comparison of before and after sanding seeds

The effect of light sandpapering on the seed coat was demonstrated in Fig. 2. Upon sandpapering, the inner layer of the seed coat became visible and appeared to be of a darker shade in comparison to the black surface layer.

Figure 2 Comparison of seeds before and after sandpaper grinding (magnification: 10 × 0.75).

Seed water absorption curve

The water uptake of caper seeds with intact seed coats increased progressively from 2 to 16 h, and stabilized at 36.99% after 16 h (Fig. 3). Conversely, the water absorption rate of caper seeds with damaged seed coats escalated gradually between 2 to 10 h and stabilized at 51.11%, a difference of 38.17% higher than that of the intact seed coat. This suggests that the seed coat abrasions lead to an increased water absorption in caper seeds.

Figure 3 Water absorption curve of caper seeds.

Effects of extracts of caper seeds on seed germination and seedling growth of Chinese cabbage

Effects of water extracts soaked with caper seeds at different concentrations and for different time periods on seed germination and seedling growth of Chinese cabbage

Table 1 highlighted the varied effect of two water extracts on the germination and growth of cabbage seeds. The treatment of cabbage seeds by four different concentrations of water extracts exhibited the strongest inhibition of germination and growth of cabbage seeds by 100% water extracts with (70.00 ± 0.09)% germination rate; this was 21.35% lower than the distilled water control; cabbage had the shortest root length and bud length. Three different time periods of the extracts whose effect on the germination and growth inhibition of cabbage seeds were as follows: extract 1 > extract 2 > extract 3.: extract 1 > extract 2 > extract 3. The lowest germination rate of cabbage seeds was (83.33 ± 0.02)% after the treatment with extract 1, which decreased by 6.37%. The inhibition of root and hypocotyl of cabbage seedlings by 75%, 100% aqueous extract and extract 1 were significantly different (P < 0.01). The inhibitory effect of 50% aqueous extract on cabbage seedling roots showed significantly difference (P < 0.05). The inhibition of cabbage seed germination by 100% water infusion extract was highly significant difference (P < 0.01). After the cabbage seeds were treated with 75% and 100% water extracts, the cotyledons turned yellow.

Table 1 Effects of aqueous extract solution of caper seeds at different concentrations and time periods on seed germination and seedling growth of Chinese cabbage.

Treatment	Chinese cabbage	
Germination rate (%)	P-value	Root length (cm)	P-value	Bud length (cm)	P-value	Cotyledon color	
Controlcheck (distilled water)	89.00 ± 0.05		2.10 ± 0.71		2.74 ± 0.69		Green	
25% water extract	86.00 ± 0.02	0.65	1.74 ± 0.92	0.21	2.53 ± 0.49	0.99	Yellow-green	
50% water extract	80.67 ± 0.01	0.18	1.62 ± 0.91	0.09	2.29 ± 0.55*	0.048	Yellow-green	
75% water extract	73.33 ± 0.11*	0.02	1.35 ± 0.68**	0.009	2.18 ± 0.45	0.99	Yellow	
100% water extract	70.00 ± 0.09**	0.005	0.86 ± 0.48**	0.00	1.82 ± 0.38	0.98	Yellow	
Extract 1	83.33 ± 0.02	0.36	1.34 ± 1.48**	0.007	1.78 ± 0.38	0.97	Yellow-green	
Extract 2	86.67 ± 0.02	0.73	1.63 ± 1.20	0.10	2.60 ± 0.54	0.99	Yellow-green	
Extract 3	90.67 ± 0.04	0.73	1.82 ± 1.68	0.33	2.77 ± 0.66	0.99	Green	
Note:

Asterisks (**, *) indicate significant at the level of 0.01 and 0.05, respectively.

Effects of different concentrations of methanol extracts of caper seeds on the germination of Chinese cabbage seeds

The results revealed a significant reduction in the germination rate of cabbage seeds following treatment with methanolic extracts of caper seeds (Table 2). It was observed that the 50% methanolic extract had the most pronounced inhibitory effect on germination, with a rate of only 4.67 ± 0.02%—a decrease of 68.17%. The inhibitory effect of both 50% and 100% methanolic extracts on cabbage seed germination was found to be statistically significant (P < 0.05). Furthermore, treatment with different concentrations of methanolic extracts resulted in yellowing of the cabbage cotyledons.

Table 2 Effects of different methanol extract of caper seeds on seed germination and seedling growth of Chinese cabbage.

Treatment		Chinese cabbage	
Germination rate (%)	P-value	Cotyledon color	
Controlcheck (80% methanol solution)	14.67 ± 0.05		Yellow	
25% methanol extract	14.00 ± 0.06	0.86	Yellow	
50% methanol extract	4.67 ± 0.02*	0.02	Yellow	
75% methanol extract	8.67 ± 0.01	0.14	Yellow	
100% methanol extract	6.00 ± 0.03*	0.04	Yellow	
Note:

An asterisk (*) indicates significant at the level of 0.05.

Germination test of caper seeds

Effects of different concentrations of FeSO4 on seed germination

the highest germination potential of caper seeds was achieved at a concentration of 7.3% FeSO4, reaching 32.00% (Fig. 4A). Conversely, the highest germination rate was observed at a concentration of 0.5% FeSO4, with a rate of 7.33%. The germination potential of seeds soaked in FeSO4 solutions ranging from 0.3% to 0.9% was found to be significantly different from the control (P < 0.01), while the germination potential of seeds soaked in 0.1% and 1.1% FeSO4 solutions showed a significant difference from the control, though to a lesser extent (P < 0.05). No significant differences were found in germination rate.

Figure 4 Effect of different chemical reagents on caper seeds germination (A), FeSO4; (B), KNO3; (C), PEG; (D), NaCl.

Asterisks (*, **) indicate significant at the level of 0.01 and 0.05, respectively.

Effects of different concentrations of KNO3 on the germination of caper seeds

As demonstrated in Fig. 4B, the highest germination potential and rate were both achieved at a concentration of 3.5% KNO3, with a potential of 29.33% and a rate of 12.00%. Seeds treated with KNO3 solutions at concentrations ranging from 3% to 4.5% showed highly significant differences in germination potential compared to the control (P < 0.01). The germination potential of seeds treated with 2% and 5% KNO3 was also found to be significantly different from the control (P < 0.05). However, no significant differences were noted in germination rate.

Effects of different concentrations of PEG on seed germination of caper

The germination potential and rate of caper seeds soaked in different concentrations of PEG were found to vary (Fig. 4C). The highest germination potential was observed at a concentration of 30% PEG, with a germination potential of 34.67%. Meanwhile, the highest germination rate was recorded at a concentration of 25% PEG, reaching 18.67%. Compared to the control, significant differences were observed in germination potential for treatments with 15%, 20%, 25%, 30%, and 35% PEG (P < 0.01), while a significant difference was observed in germination potential for the treatment with 10% PEG (P < 0.05). Additionally, a highly significant difference was observed in the germination rate for the 25% PEG treatment (P < 0.01).

Effects of different concentrations of NaCl on seed germination

The germination potential of caper seeds was found to reach 30.00% at a NaCl concentration of 50 mmol/L and the germination rate was highest, at 19.33%, when the NaCl concentration was 25 mmol/L (Fig. 4D). These results indicate that there were very significant differences in germination potential (P < 0.01) when compared to the control group at a NaCl concentration of 50 mmol/L, while the germination rate was found to be highly significantly different (P < 0.01) at a concentration of 25 mmol/L.

Effects of different concentrations of GA3 on seed germination

Caper seed germination was positively influenced by increasing concentrations of GA3, with a maximum germination potential of 30.67% observed at a concentration of 200 mg/L and a maximum germination rate of 19.33% observed at a concentration of 250 mg/L (Fig. 5). Compared to the control, there was a very significant difference in germination potential (P < 0.01) when the GA3 concentration was 200 mg/L, while concentrations of 175 and 225 mg/L showed significant differences in germination potential (P < 0.05). Significant differences in germination rate (P < 0.05) were also observed for GA3 concentrations of 225 and 250 mg/L.

Figure 5 Effects of different concentrations of GA3 on seed germination of caper.

Asterisks (*, **) indicate significant at the level of 0.01 and 0.05, respectively.

Effects of different days in sand storage on seed germination

The results of the sand storage treatment on caper seed germination rate and potential showed a gradual increase over time (Fig. 6). Compared with room temperature, the treatment effect of 4 °C was better. The optimal treatment conditions were determined to be a storage time of 21 days at 4 °C, which resulted in the highest germination rate (17.67%) and potential (26.67%) observed. Significant differences in germination rate and potential were observed between the control and treatment conditions of 14 and 21 days at 4 °C (P < 0.01). The germination potential also showed a significant difference at these conditions (P < 0.05).

Figure 6 Effects of different sand storage time and temperature on seed germination of caper.

Asterisks (*, **) indicate significant at the level of 0.01 and 0.05, respectively.

Discussion

Morphological anatomical observations of caper seeds have revealed instances of seed abortion, which negatively impacts seed germination (Willis et al., 2014). There are numerous factors that influence seed germination, including the thickness and structure of the seed coat (Greipsson, 2001; Zhang, 2021). A dense seed coat structure with well-developed palisade tissue can result in poor water and air permeability, causing seed dormancy (Vazquez-Yanes, 1976; Zhu et al., 2022). Results from the caper seed water absorption experiment showed that grinding the seed coat increased the water absorption rate by 38.17% compared to the unground seed coat, highlighting the presence of a water absorption barrier in the seed coat. Lin et al. (2016) found that the water uptake rate of caper seeds in the group with the digestive tract of Teratoscincus scincus was consistently higher than the control group, indicating that the digestive tract of Teratoscincus scincus promotes seed water uptake. One of the primary mechanisms through which fruit-eating animals enhance seed germination is the abrasive and corrosive effects of the digestive tract on the seed coat, which enhances seed permeability to water and gas. Xiao et al. (2017) investigated whether the digestive tract of Hemiechinus auritus impacted the water uptake and germination of Capparis spinosa seeds and found that the digestive tract increased the seed’s water uptake rate and absorption capacity. The germination rate of seeds soaked in GA3 solution and with the digestive tract of H. auritus was significantly higher than the control group, with extended germination days, suggesting that the digestive tract of H. auritus improves seed permeability to water and disrupts mechanical obstructions, promoting seed germination. This result aligns with the present study’s findings, indicating that the caper seed coat does affect its germination rate.

It has been established that the presence of germination inhibitors in seeds can contribute to seed dormancy (Crocker, 1916). The existence of physiological barriers, particularly in fully developed seed embryos, further complicates the germination process (Gou, Shen & Shi, 2019). The study of germination inhibitors in caper seeds revealed that the aqueous extract of seed powder had a more pronounced inhibitory effect on the growth of cabbage seedlings, particularly in regards to root growth, indicating the presence of germination inhibitors in caper seeds. This result was confirmed by the observation that high concentrations of the aqueous extract could lead to yellowing of the cotyledons of cabbage seedlings, potentially due to inhibition of chlorophyll synthesis.

The results of our study indicate that, compared to the aqueous extract, the methanolic extract displayed a more potent inhibitory effect on the germination and growth of cabbage seeds. Our findings suggest that the germination inhibitor present in caper is more soluble in methanolic solution, as evidenced by the lowest germination rate observed in cabbage seeds treated with the methanolic extract. This is consistent with the findings of Sun & Ji (2009), who found that extracts with varying concentrations of 80% methanol had a greater impact on the germination rate of Tilia seeds compared to water extracts. Our findings suggest that the germination inhibitory substances in the methanolic solution are higher compared to other extracts. This is in contrast to the findings of Shao et al. (2018), who found that the water extract of Polygonatum seeds exhibited higher inhibitory activity, while the ethyl acetate extract had no significant impact on cabbage seedling germination. These results suggest that the germination inhibitor in Polygonatum sibiricum seeds is water-soluble. In this experiment, two controls were established by soaking cabbage seeds in distilled water and 80% methanol solution, with the latter showing a significantly lower germination rate (14.67 ± 0.05) of 83.52% compared to the distilled water control. The inhibitory effects of methanol on seed germination have been well documented, with studies such as those by Zhang (2015) and Meng, Ji & Shao (2017) indicating that aging maize and cotton seeds with 50% methanol respectively, resulted in reduced germination rates and seedling growth. It is speculated that the methanolic extract of caper seeds exhibits a significant inhibitory effect on the germination of cabbage seeds due to the combination of its effects on seed aging and the greater solubility of the germination inhibitor in methanol.

In this study, three methods including the use of chemical reagents, exogenous hormones, and sand storage were employed to treat caper seeds. The highest germination potential (35%) was observed when the caper seeds were soaked in 30% PEG, while the highest germination rate (19%) was achieved when the seeds were treated with 250 mg/L GA3 and 25 mmol/L NaCl. Previous reports have indicated that soaking caper seeds in GA3 can significantly enhance the germination rate over an extended period. When sulfuric acid was combined with GA3, the highest germination rate (60%) was obtained through the sequential application of a 30-min sulfric acid soak followed by GA3 solution immersion (Sottile et al., 2021). Zhang et al. (2009) investigated methods to improve the germination rate of prickly mandarin, and the highest germination rate of caper seeds (16%) was recorded when the concentration of PEG reached 25%. Sun & Ma (2010) found that the highest germination rate (700 mg/L GA3) was achieved by soaking the seeds in concentrated H2SO4 for 70 min and treating them with H2O2 for 4 h. In the present study, however, the highest germination rate observed was only 19% after GA3 treatment. Such variations could be attributed to the different origins of the seeds and the environmental factors, including climate and genetic variations, at their place of origin. The germination rate and potential of caper seeds exhibited a trend of first increasing and then decreasing with the soaking in varying concentrations of NaCl solution, demonstrating the salt tolerance of these seeds. Sadeghi & Rostami (2016) investigated the effect of salt stress on prickly mountain citrus and found no significant decrease in the fresh weight of the plants or in root tolerance under salt stress. Furthermore, the synthesis of abscisic acid was observed to increase with increasing salinity. This plant has been reported to grow well in poor soils and is known to thrive in sandy soils with low organic matter content, and has a strong resistance to salinity (Yazdani Biouki, Khajahhosseini & Rad, 2021).

During the germination of caper seeds, a high incidence of mold was observed, which may have contributed to the low germination rate recorded in this study. Lian et al. (2021) showed that pepper seeds scalded for 15 min at temperatures below 50 °C had a high rate of mildew and poor disinfection. On the other hand, scalding at temperatures above 50 °C resulted in a significant reduction in both germination rate and potential. In a study conducted by Yuan (2012), the impact of different scalding conditions on Mungbean sprout growth was evaluated through measurement of water absorption rate, germination rate, and sprouting bean mass ratio. The results indicated that scalding at 55 °C for 20–30 min produced the optimal results, with a germination rate of 100% and a sprouting bean mass ratio of over 4.81, leading to optimal sprout growth. It is worth noting that caper seeds have a thick seed coat, and scalding can be utilized to reduce the occurrence of mold and improve germination rate.

Conclusions

This study provided initial evidence of the presence of seed germination inhibitors in caper seeds. The extracts from caper seeds exhibited an inhibitory effect on the germination of cabbage seeds, with methanolic extracts demonstrating the strongest inhibition. The maximum germination rate of 19.33% was observed in seeds treated with a 250 mg/L GA3 and 25 mmol/L NaCl solution. Further investigations into the structural identification and content analysis of these germination inhibitors would shed light on the underlying mechanism of seed dormancy in capers, offering a theoretical foundation for future research.

Supplemental Information

Supplemental Information 1 Original data.

Click here for additional data file.

Thanks to the tutor (Liping Xu, Yili Normal University) for providing helpful comments on the manuscript and Huagui Qi (Yili Normal University) for modifying the language of the article.

Additional Information and Declarations

Competing Interests

Author Contributions

Data Availability

The authors declare that they have no competing interests.

Min Wang conceived and designed the experiments, performed the experiments, analyzed the data, prepared figures and/or tables, authored or reviewed drafts of the article, and approved the final draft.

Xiaolu Yuan performed the experiments, authored or reviewed drafts of the article, and approved the final draft.

Liping Xu conceived and designed the experiments, authored or reviewed drafts of the article, and approved the final draft.

The following information was supplied regarding data availability:

The raw data is available in the Supplemental Files.

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
