# Peer review of "Preliminary study on Bioassay of Capparis spinosa L. seed extract and seed germination"

_PeerJ, doi:10.7717/peerj.15082_

## Round 0.1 · original submission · Major Revisions

Dear Authors

Kindly see the comments of all reviewers. Kindly include their all suggestions before re-submission. Thank you.

Editor

·

Basic reporting

The manuscript (Preliminary study on characteristics and release of seed dormancy of Capparis spinosa L.) presents the characterization of Capparis spinosa L seed dormancy. The manuscript is poorly drafted and have many grammatical and language flaws. For instance, present tense is used in abstract and material and methods and results section. The material and methods and results are mostly written in past tense. Partitioning Abstract in Objectives, Methods, and Results is unnecessary. The abstract section must be re-written thoroughly following the style of any latest paper published in PeerJ.
The information written in the introduction is not well connected and coherent. Please include economic importance of this species in the first paragraph and improve the language in the Intriduction section. Must rewrite the following sentences to improve the readability. “In this experiment, the germination inhibitors of caper are preliminarily discussed, and 76 FeSO4, KNO3, PEG-6000, NaCl, GA3 soaking and sand storage are used to treat seeds to break 77 the dormancy of seeds and promote and improve the germination rate of seeds. The sustainable 78 development and utilization of this plant resource provides scientific basis.”

Experimental design

Material and Methods

Line 81-82: “Materials are collected at the end of July 2021 in Karayagaqi Township, Yining County
(44.10°N, 81.52°E, altitude 1101.37m)”. Please specify the materials and use correct language for writing the Material and Methods section.

The language of material and methods is very poor and it is very hard to grasp the sense. Please add/edit the following information to material and methods section besides improving language
-How many seeds were used to observe the shape of the seed section under a stereo microscope?
-Remove the experimental apparatus section and mention the instruments wherever used in the experiments.
-Add the fourmuli of calculating absorption rate
-Use the equation editor option to write statistical formula
-Properly cite the software . SPSS Statistics 26

Validity of the findings

Results
Re-write all results in past tense. The language used throught the Results section is incorrect.
Reconstruct Table 1 and 2. Add a column for P vale and remove the rows used for p values

Figures 1 and 2 are not of scientific quality. Improve the resolution of these figures
Figures 5- 10 are hardly readable. These figures can be combined to 2-3 figures.
Discussion
Discussion section is very poor. Besides language errors the results are not discussed logically

Reviewer 2 ·

Basic reporting

Title: Preliminary study on characteristics and release of seed dormancy of Capparis spinosa L.
Q1: The title is unclear. It must be reviewed so that it reflects the importance of the work, which lies among other things in the effect of the extract of the seeds of the caper on the germination of the seeds of Chinese Cabbage.
Abstract:
Objective: The objective of the present study to explore the characteristics of caper seed germination inhibitors, and to explore the methods to eliminate or alleviate the seed germination inhibition.
Q1: You have studied the effect of methanolic and aqueous extracts of the caper bush on the germination of Chinese cabbage seeds, you must also mention it.
The cross-sectional structure of caper seeds is observed by microscope; the changes of water absorption are observed after the seeds are ground with and without the seed coat respectively; different kinds of extracts of caper seeds and the germination of Chinese cabbage seeds are studied. Seedling growth; the methods of chemical reagents, hormone soaking, and sand storage are used to improve the germination rate of caper seeds.
Q2: This paragraph should be reformulated. It is not clear. The description of the methodology is very limited and does not reflect all the methods adopted.
Introduction:
Q1: You must also give an overview of the germination of Chinese Cabbage seeds and the factors that affect it.

Experimental design

Materials and methods:
Line 83:
Q1 : Was the collection made from the same genotype? or from different genotypes?
Q2 : It is a spontaneous plantation or a domesticated field ?
Line 95 :
Q1 : What are the grain storage conditions after harvest?

Line 99 :
Q1: This is the effect of the aqueous and methanolic extract of caper seeds on the germination of cabbage seeds. Therefore, this title must reflect the nature of the extracts adopted.
Line 104 :
Q1: How many petri dishes are used in each repetition?
Line 114:
Q1: What do you mean by 30 groups??
Line 121-124:
Q1: the choice of concentrations must be justified.
Line 124:
Q1: 50 seeds per treatment*3, right?
Line 133-136:
Q1: The humidity rate and the temperature of the sand were not measured during the test ?
Line 136-139:
Q1: It is need to be rephrased, the meaning is not clear.

Validity of the findings

Results
Line 176-186:
Q1: You must homogenize the presentation of the results. In the materials and methods part, you announced the methanolic extract first but in the results section you started with the aqueous extract.
Q2: The results are interesting but it is necessary to reformulate the text in order to make it fluid for the reader.
Line 222-223:
Q1: Is it FeSO4 or PEG ?
Discussion: This section should be reformulated.
Line 253-258:
Q1: This part needs to be further developed and fed with more references
Line 271 :
Q1 : You must remember the difference between the two extracts.

·

Basic reporting

Thank you for submitting your manuscript "Preliminary study on characteristics and release of seed dormancy of Capparis spinosa L." to PeerJ. Following careful consideration by the journal's instructions, I regret to inform you that your manuscript does not meet journal's requirement, primarily because of very poor drafting of the manuscript and very low quality images.

Experimental design

Information regarding experimental design in clearly missing.

Validity of the findings

No comment

Reviewer 4 ·

Basic reporting

Study the characteristics and release of seed dormancy of Capparis spinosa L. is of great importance for improving seed germination rate of the plant. In the study, seed water absorption, the inhibitory effects of water and methanol extracts of capper seeds on the germination of Chinese cabbage seeds, and the germination potential and germination rate of capper after treated with FeSO4, PEG, GA3, NaCl were tested. The experiments and the data analysis were too simple to support the conclusions and lack of novelty.
1、 In previous study as mentioned in the manuscript, it was found that after soaking the seeds with concentrated H2SO4 for 70 min, treated with H2O2 for 4h, and then treated with 700mg/L GA3, the germination rate of capers reached 68%. But In this study, the highest germination rate after treatment with GA3 was only 19%. What’s the function of H2SO4 and H2O2 for capper seed germination? Why not design experiments on this basis to improve seed germination rate?
2、 The experiments for testing the inhibitory effects of water and methanol extracts of capper seeds on the germination of Chinese cabbage seeds are not very rigorous. The extracts may be harmful for the growth of Chinese cabbage, not only effect the germination of it. Did the authors test the inhibitory effects of the extracts of capper seeds on its own germination rate?

Experimental design

The experiments and the data analysis were too simple to support the conclusions and lack of novelty.

Validity of the findings

The findings is lack of scientific value and application value.

---

## Round 0.2 · Major Revisions

The authors need to include all suggestions given by reviewers before re-submission.

·

Basic reporting

Dear Editor

I am already have reviewed this manuscript and recommended the rejection due to very low experimental, presentation and writing quality. I wonder why I received the same manuscript after the final recommendation as rejected.

Respecting the editor's request, I still performed a review and observed the following mistakes or confusing sentences.
Abstract
-. In our experiments, we investigated the eûects of diûerent types of infusions of caper seeds on the germination and seedling growth of cabbage seeds.
-What is the difference between caper seed and Chinese cabbage? Why the authors don't use a consistent terminology?
-Even in revised version the writing quality is very poor and results description is confusing. For example read the following sentence in the abstract
"From the results of the germination
experiment of caper seeds, it is found that when the seeds were soaked with 30% PEG, the
germination potential was the highest, which was 35.00%; When the seeds were soaked in
250 mg/L GA3
and 25 mmol/L NaCl, respectively, the germination rate was the highest,
which was 19.33%."
Authors are giving confusing statements about the highest value of the germination.
-The study is of preliminary level as mentioned in the title and the experimental data and evidences are not sufficient to claim it as of industrial scale. The following conclusion statement is nothing but an exaggeration.
"The research on the germination-inhibiting substances in
seeds is an applied research closely related to industrial and agricultural production, which
is of great signiûcance for the large-scale artiûcial breeding of capers can be realized."

Materials & Methods
-All equations must be written usiy equation editor
-The following heading in the material and methods makes no sense
"The aqueous and methanolic extract of caper seeds on the germination of cabbage
sseeds"
"Seeds are treated by the method of sand storage"
-The statey is confusing "Compared with the control, to
162 determine treatment and interaction effects, analysis of variance (ANOVA) was carried out on
163 the root length, shoot length, germination percentage of Chinese cabbage seeds, and the
164 germination percentage and germination potential of caper seeds."

The presentation of results and their discussion needs still improvements. The language should be simple and free of errors.

Experimental design

The experiment design is satisfactory but the description is not clear and satisfactory

Validity of the findings

The validity and findings are of very low significance. The depth of the study is not enough to claim it as a significant contribution to the industry or agriculture sector.

Additional comments

I still think that this manuscript present data and results of low significance. Regarding th quality standards of PeerJ, I think this manuscript doesn't qualify the acceptance.

Reviewer 2 ·

Basic reporting

After reading the corrections made by the authors, I think that this work deserves to be published.

Experimental design

After reading the corrections made by the authors, I think that this work deserves to be published.

Validity of the findings

After reading the corrections made by the authors, I think that this work deserves to be published.

---

## Round 0.3 · accepted · Accept

The authors have tried to include reviewers' suggestions.